# Disease–Gene Networks of Skin Pigmentation Disorders and Reconstruction of Protein–Protein Interaction Networks

**DOI:** 10.3390/bioengineering10010013

**Published:** 2022-12-21

**Authors:** Cian D’Arcy, Olivia Bass, Philipp Junk, Thomas Sevrin, Giorgio Oliviero, Kieran Wynne, Melinda Halasz, Christina Kiel

**Affiliations:** 1Systems Biology Ireland and UCD Charles Institute of Dermatology, School of Medicine, University College Dublin, Belfield, D04 V1W8 Dublin, Ireland; 2Systems Biology Ireland, School of Medicine, University College Dublin, Belfield, D04 V1W8 Dublin, Ireland; 3Systems Biology Ireland, School of Medicine, and Conway Institute of Biomolecular & Biomedical Research, University College Dublin, Belfield, D04 V1W8 Dublin, Ireland; 4Department of Molecular Medicine, University of Pavia, 27100 Pavia, Italy

**Keywords:** disease–gene network, pigmentation, melanin biosynthesis, systems medicine, protein–protein interaction network, melanocytes

## Abstract

Melanin, a light and free radical absorbing pigment, is produced in melanocyte cells that are found in skin, but also in hair follicles, eyes, the inner ear, heart, brain and other organs. Melanin synthesis is the result of a complex network of signaling and metabolic reactions. It therefore comes as no surprise that mutations in many of the genes involved are associated with various types of pigmentation diseases and phenotypes (‘pigmentation genes’). Here, we used bioinformatics tools to first reconstruct gene-disease/phenotype associations for all pigmentation genes. Next, we reconstructed protein–protein interaction (PPI) networks centered around pigmentation gene products (‘pigmentation proteins’) and supplemented the PPI networks with protein expression information obtained by mass spectrometry in a panel of melanoma cell lines (both pigment producing and non-pigment producing cells). The analysis provides a systems network representation of all genes/ proteins centered around pigmentation and melanin biosynthesis pathways (‘pigmentation network map’). Our work will enable the pigmentation research community to experimentally test new hypothesis arising from the pigmentation network map and to identify new targets for drug discovery.

## 1. Introduction

Melanin is a photo-protectant and oxidation neutralizing pigment derived from the melanocyte cells in the basal membrane of the epidermis and helps maintain skin homeostasis [1]. Melanin is a polymer synthesized in melanosomes (as some of the intermediates in melanin synthesis are toxic to the cell), a lysosomal-like organelle located within melanocytes. This process, melanogenesis, is a complex sequence of biochemical reactions starting with the oxidation of L-tyrosine. It is regulated by both intrinsic (genetic) and extrinsic (UV radiation) factors as well as in response to melanocytic hormones such as adrenocorticotropin, estrogens and progesterone which influence pigment production [2].

Melanocytes are neural crest derived cells. Depending on their migration from the neural crest, they can land in the epidermis, hair follicle, eye, inner ear, bones, heart, and/or brain [3,4]. Cutaneous melanocytes can be described as classical melanocytes and those found elsewhere in the body as non-classical melanocytes. Melanin, alongside hemoglobin, carotene and bilin, is the main contributor to human pigmentation in hair, skin, and eyes [5]. Location is vital to acknowledge as it may suggest a co-morbidity association with pigmentation diseases. In the epidermis melanocyte homeostasis is maintained by the contact with the surrounding keratinocytes at a ratio of around one melanocyte to 35 keratinocytes [6]. This is known as the epidermal melanin unit (EMU), in which the keratinocytes regulate melanocyte growth and proliferation. In response to UV the keratinocytes release α-melanocyte-stimulating hormone (α-MSH) which induces the production of melanin in the melanocytes [2]. The melanosomes are then transported through the dendritic-like protrusions of the melanocytes around the keratinocyte and are released [7]. Melanin then accumulates over the nucleus of the keratinocyte protecting it from UV [2].

Two main forms of melanin are found in epidermal melanocytes: eumelanin and pheomelanin [2]. As a result of the differing ratio between these two melanin pigments, as well as the amount of melanin in each melanosome, different skin/skin appendage pigments are seen in humans. Eumelanin, a polymer of 5,6-dihydroxyindole (DHI), and 5,6-dihydroxyindole-2-carboxylic acid (DHICA), has black to brown pigments whilst pheomelanin, a polymer of benzothiazine intermediates, gives pigments in a range of red and yellow. In conjunction with pigment production the major functions of melanin in humans are the protection from UV radiation, thermoregulation and in some studies it is linked to antibiotic properties [2,8]. The EMU provides photo protection, as eumelanin especially is an excellent absorber of incoming energy from the UV radiation and dissipating it into heat [2]. Thus, melanin provides an essential protective role in the epidermis not only from UV radiation but also the neutralization of oxidative stress and possible antibiotic functions. It is therefore likely that disruption to the biochemical role of melanin production in other parts of the body may have a role in disease.

In the epidermis, melanin biosynthesis occurs within the melanosome organelle of the melanocyte. This process is controlled by a combination of enzymatic and chemical reactions, leading to the eventual production of eumelanin or pheomelanin [9]. Reactions resulting in pigmentation typically begin with exposure of the keratinocyte of the epidermis with UV radiation or a pollutant. This leads to the production of the proopiomelanocortin (POMC), a precursor to α-MSH, from the keratinocyte cells. α -MSH will then travel to the adjacent melanocytes and bind to MC1R receptor. This binding activates adenylate cyclase which will convert ATP into cyclic adenosine monophosphate (cAMP) within the melanocyte. This increase in intracellular cAMP up regulates transcriptional factors such as CREB and microphthalmia-associated transcription factor (MITF) [10]. MITF drives production of enzymes such as dopachrome tautomerase (TYRP2), tyrosinase (TYR) and tyrosine-related protein 1 (TYRP1) leading to melanogenesis [11].

Raper–Mason pathway leads to the formation of eumelanin and pheomelanin in humans [12,13]. Formation begins initially with the transformation of L-tyrosine into L-dopaquinone catalyzed by TYR. Addition of L-cysteine will induce formation of benzothiazine units leading to the product of pheomelanin. Where there is absence of L-cysteine, formation of eumelanin, formally known as eumelanogensis, occurs and L-dopaquinone is spontaneously transformed into L-leukodopachrome from lack of L-cysteine. L-leukodopachrome reacts with L-dopaquinone to yield L-dopachrome. L-dopachrome, in accordance with the levels of TYRP2 and decarboxylation rates, will either be converted into DHI or DHICA. Further oxidation of DHI will yield indoquinones, which are important in polymerization for final eumelanin product. Cross-linkage and polymerization of DHI, DHICA and indoquinone give rise to eumelanin [2]. The determining factor of whether the pheomelanin or eumelanin pathway will be initiated is due to pH, intracellular L-cysteine levels and Slc7al which is a gene encoding for cysteine transporter [14,15]. Pheomelanin and eumelanin can result in different characteristics. For example, individuals with more pheomelanin may be more prone to blistering or carcinomas due to its increase in reactive oxygen species accelerating carcinogenesis [16].

To appreciate how pigmentation disorders occur requires understanding the heterogeneity, development, regeneration and senescence of melanocytes and their precursors. All enzymes, proteins, and genes in the biosynthesis of melanin can be targets of mutation which can lead to clinical presentation of disease. Diseases in pigmentation can be classified into either hypo pigmented, hyper pigmented or mixed hyper and hypo pigmented [17]. These can then be further divided into congenital and acquired disease. An example of congenital hyperpigmentation is Carney Complex, an autosomal dominant disorder which is described clinically as a ‘complex of myxomas, spotty pigmentation and endocrine overactivity’ [18]. An example of an acquired hyperpigmentation disorder is melasma. This is a very harmless disorder which appears clinically as brown macule in the malar region of women. This is a disorder that has an indirect effect on pigmentation through a defect in the cAMP-dependent protein kinase type I-alpha regulatory subunit (PRKAR1A) gene which leads to an increase in cAMP and increasing pigmentation via upregulation of MITF [19]. Oculocutaneous albinism is a congenital hypopigmentation disorder, where absence or reduction in melanin biosynthesis occurs due to, for example, lack of the enzyme tyrosinase [20]. Albinism manifests as either syndromic or non-syndromic phenotypes. Loss of tyrosine can gradually occur throughout life because of oxidative posttranslational modifications [21], clinically showing a gradual increase in albinism with age. An example for hereditary hypopigmentation disorder is the Hermansky–Pudlak syndrome which leads to hypopigmentation due the mutation of proteins related to the transport of melanosomes in the skin. This also affects the transport of lysosome related organelles in different organs [22].

Network-centric approaches in medicine are powerful means to achieve a comprehensive understanding of the molecular basis underlying the disease and to investigate how the impact of genetic defects propagates across the network [23,24]. Indeed, in the disease module hypothesis, a group of connected genes that represent a disease module often share similar cellular functional roles [25]. This can be seen in many disorders and cancers; it is however quite evident in monogenetic diseases that can be caused by different genes that are often part of the same pathway [26]. Investigating interlinked genes between diseases will allow for a better understanding of the phenotypic presentation of the disease and offer potentially novel therapeutic targets, disease prevention and diagnostic approaches [27,28,29]. As seen in recent publications there are diseases such as retinal dystrophies (RD) with varying subsets, and complexities [26]. Although these subsets produce the same pathophenotype, their genetic networks do not overlap. For example, genes involved in monogenic RD affect core vision related functions (e.g., phototransduction, retinal recycling), while genes involved in complex and age-related RD are linked to general tissue homeostasis processes (e.g., extracellular matrix remodeling, inflammation) [26].

Here, we obtained gene-disease networks for pigmentation-related disorders. Using protein–protein interaction (PPI) information we reconstructed networks centered around pigmentation gene products (‘pigmentation proteins’), which were filtered for expression in a panel of both pigmented and unpigmented melanoma cell lines. Altogether, our work provides a systems representation of PPIs centered around pigmentation in melanocytes (‘pigmentation network map’).

## 2. Materials and Methods

### 2.1. List of Genes Linked to Pigmentation and Pigmentation Disorders

To assemble a list of human genes linked to pigmentation and pigmentation disorders, we obtained a list of genes involved in pigmentation disorders from a recent review article [17] (here called ‘Yamaguchi gene list’) (Appendix A). Gene symbols were matched to official gene names of 19300 protein-coding genes [30] and unmatched genes were removed. For all pigmentation disorder listed in [17], each disorder name was used as a search name in the OpenTargets platform (https://www.opentargets.org/; accessed on 13 July 2022) to obtain additional genetic association linked to the respective disorder by filtering on evidence ‘Genetic Associations’ (here called ‘OpenTargets gene list’). Gene symbols were matched to official HGNC names [30] and unmatched genes were removed. Furthermore, genes linked to pigmentation were obtained from a recent publication by Baxter et al. [31] (Appendix A of that publications, only human phenotypes; here called: ‘Baxter Appendix A gene list’). Gene symbols were matched [30] and unmatched genes were removed. In total, 278 gene-disease associations were obtained. Phenotype information was obtained from the OMIM database (https://www.omim.org/; accessed on 4 August 2022) (Appendix A). Based on the phenotype information, we classified phenotypes into four groups: (1) Hyperpigmentation disorder, (2) Hypopigmentation disorder, (3) Mixed hypo- and hyperpigmentation disorder, and (4) pigmentation phenotype without disease. In addition to the table containing all 278 gene-disease associations, two separate tables containing either only all 243 genes (Appendix A) or only all 174 diseases/ phenotypes (Appendix A) without redundancy were generated.

### 2.2. Protein–Protein Interactions among Pigmentation Proteins and Clustering Using STRING

To obtain protein interactions among the 243 pigmentation proteins, the STRING database [32] was used and interactions from all active interaction sources and a minimum required interaction score of 0.700 high confidence were obtained (Appendix A). Network clusters were obtained in STRING using the Markov Cluster (MCL) algorithm with an inflation parameter at 2.5. Interactions and MCL cluster information were exported to Cytoscape software version 3.9.1.

### 2.3. Gene Ontologies Using the Sysgo Database

The SysGO database contains (main) functional annotation for each protein-coding gene (321 classes; “SysGO—set 1”), of which 132 are related to signaling functions [30]. For visualization purposes, some classes are merged into a total of 58 groups (“SysGO—set 2”), which is further simplified to 15 groups “SysGO—set 3”) of the classes “Signaling”, “Metabolism”, “Protein translation, folding, modification and degradation”, “Transcription”, “Unknown”, “Cytoskeleton”, “Organelles”, “Other”, “Immune system and Inflammation”, “Chromatin organization and DNA repair”, “Neuronal System, synapses, channels”, “ECM organization”, “Cell junction and adhesion”, “Developmental”, and “DNA Replication”.

### 2.4. Gene Set Enrichment Analysis

GSEA for networks was carried out using the gprofiler2 software [33]. A significance threshold of 0.05 was applied using the Benjamini–Hochberg FDR. The following data sources were used: molecular function (MF), biological process (BP), wikipathways (WP) and human phenotype (HP).

### 2.5. Tissue Culture

A375 (unpigmented) and FM55 (pigmented; with BRAFV600E mutations) melanoma cell lines were received from the McCann group (Dublin, Ireland) and the Tobin group (Dublin, Ireland), respectively. All cells were cultured in Dulbecco’s modified Eagle medium (DMEM) high glucose (4500 mg/L glucose, 0.2 mM L-glutamine, Thermo Fisher Scientific; Waltham, MA, USA), 10% (*v*/*v*) heat inactivated fetal bovine serum (FBS) (heat inactivated, Thermo Fisher Scientific), and Penicillin-Streptomycin (Gibco; Waltham, MA, USA). All cells were cultured at 37 °C and 5% CO_2_ in a humidified incubator.

### 2.6. Whole-Cell Mass Spectrometry Analysis of Melanoma Cell Lines

All melanoma cells were cultured in triplicate in 6 well-plates, at a seeding density of 3 × 10^5^ cells/well. All cells were lysed in radioimmunoprecipitation assay (RIPA) lysis buffer, which also contained complete protease inhibitor cocktail tablets (Sigma; St. Louis, MO, USA). Following centrifugation at high speed (14,000 rpm) for 10 min at 4 °C the supernatant was collected in a new tube. Using the PierceTM BCA protein assay (Thermo Fisher Scientific) the protein concentration of cell lysates was determined, following the manufacturer protocol. All samples, were adjusted in 50 µL of lysis buffer to contain approximately 50 µg of proteins. The same volume of urea (8 M) was added (50 μL), followed by 100 μL of 100 mM ammonium bicarbonate (NH_4_HCO_3_) and 10 μL of 100 mM of calcium chloride (CaCl_2_). The protein samples were reduced in 0.2 M of dithiothreitol (DTT) for 15 min at room temperature. Subsequently, the samples were adjusted with 0.4 M of iodoacetamide (IAA) to a final concentration of 4 mM and incubated for 15 min at room temperature in the dark. To prepare the magnetic beads, a combination of 5 µL of hydrophilic and 5 µL of hydrophobic beads and were mixed and washed several times in MS grade water. The beads were added to a deepwell plate in the KingFischer Duo Prime, as well as the cell lysate and the Trypsin (0.5 µg/µL). In other wells, 80% ethanol was added for the washing steps. Following overnight preparation on the KingFischer (8 h digestion and 4 °C storage), samples were transferred into new tubes with 0.1% formic acid and samples were dried in speed vacuum and stored at −20 °C until use.

### 2.7. Liquid Chromatograhy Combined with Tandem Mass Spectrometry (LC-MS/MS)

Samples were run on a Bruker timsTOF Pro mass spectrometer connected to a Bruker nanoElute nano-lc chromatography system [34]. Tryptic peptides were resuspended in 0.1% formic acid. Each sample was loaded onto Acclaim PepMap C18 trap cartridge (0.3 mm inside diameter, 5 mm length) (Thermo Scientific) and then separated on an Aurora UHPLC column (25 cm × 75 μm ID, C18, 1.6 μm) (Ionopticks) with an increasing acetonitrile gradient over 60 min at a flow rate of 300 nL/min [35].

The mass spectrometer was operated in positive ion mode with a capillary voltage of 1650 V, dry gas flow of 3 L/min and a dry temperature of 180 °C. All data were acquired with the instrument operating in trapped ion mobility spectrometry (TIMS) mode. Trapped ions were selected for MS/<X using parallel accumulation serial fragmentation (PASEF). A scan range of (100–1700 *m*/*z*) was performed at a rate of 5 PASEF MS/MS frames to 1 MS scan with a cycle time of 1.03 s [36].

The chromatography buffers used were buffer B (99.9% acetonitrile, 0.1% formic acid) and buffer A (99.9% water, 0.1% formic acid). All solvents are LC-MS grade.

### 2.8. Proteomics Data Analysis Using Maxquant

The mass spectrometry raw data were searched against the Homo sapiens subset of the Uniprot Swissprot database (reviewed) using the search engine MaxQuant (release 2.0.2.0 and 2.0.3.0) using specific parameters for trapped ion mobility spectra data dependent acquisition (TIMS DDA). Each peptide used for protein identification met specific MaxQuant parameters, i.e., only peptide scores that corresponded to a false discovery rate (FDR) of 0.01 were accepted from the MaxQuant database search. The normalized protein intensity of each identified protein was used for label free quantitation (LFQ) [37].

### 2.9. Expanded Pigmentation Network Using STRING

The pigment disorder gene list searched against the STRING network database for expansion, with a high confidence of 0.7. The ‘More’ button in the STRING interface was used to expand the network until it reached >2000 expected edges (4668 PPI). This file was downloaded and uploaded to Cytoscape and a new expanded network was produced.

## 3. Results

Disease causing gene products (proteins) often work together in cells and tissues by forming complexes, larger assemblies (‘networks’), or by functioning in the same pathway. Therefore, network-centric approaches are powerful to obtain a thorough understanding of molecular disease mechanisms [23,27]. Indeed, according to the network centric path in molecular medicine, the disease phenotype is considered as a direct result of changes of normal (physiological) intra- and intercellular networks.

Here, we used a network-centric approach to obtain a quantitative and systems view of the genes and molecular PPI networks underlying pigmentation disorders.

### 3.1. Disease–Gene Network Related to Pigmentation

To generate disease–gene network (‘diseasome’ [25,27]), using manually curated publications [17,31] and the OpenTargets database (see Methods), we obtained 278 gene-disease associations containing 243 genes and 174 pigmentation diseases/phenotypes (Appendix A). The large number of diseases partly arises from each disease subtype being considered a separate disease (e.g., Hermansky–Pudlak syndrome type 1, type 2, …). Hence, for some visualization and analyses purposes, the 174 diseases/phenotypes were further sub-classified into hyperpigmentation disorders (total of 73), hypopigmentation disorders (total of 55), mixed hyper-/hypo pigmentation disorders (total of 40), and general skin pigmentation phenotypes (total of 6).

The associations between genes and diseases were represented in form of a network (‘pigmentation diseasome’) (Figure 1 and Appendix A). The network of the 174 pigmentation diseases shows that generally only a small number of genes are shared between multiple diseases, and if they are shared, they tend to be linked to diseases belonging to the same subtype. Only a handful of genes are linked to diseases belonging to different subtypes. Examples for genes linked to multiple subtypes are FANCA (a DNA repair protein and linked to Dermal melanocytosis, Vitiligo, and Fanconi anemia, complementation group A), IRF4 (an immune-specific transcriptional activator and linked to Dermal melanocytosis, Vitiligo, and variation in Skin/hair/eye pigmentation), KIT (a stem cell growth factor receptor and linked to Mastocytosis and Piebaldism), and TYR (the key enzyme in melanin synthesis and linked to susceptibility to cutaneous malignant melanoma, Oculocutaneous albinism, type 1A and 1B, Piebaldism, Vitiligo, and Skin/hair/eye pigmentation).

### 3.2. Functional Analysis of Genes Linked to Skin Pigmentation

To analyze the functions of genes associated with pigment disorders, all genes were searched against several ontology sources using the gProfiler2 database (Appendix A). As expected, the top associated functions are abnormalities to skin pigmentation, pigment cell development and proliferation, but also immune system functions and DNA repair processes.

For a more fine-grained functional analysis, the SysGO database was used to assign each pigmentation gene to a unique function using the highest level of classification with 15 classes (see Methods and Appendix A). Comparing the 68 genes that are exclusively linked to hyperpigmentation diseases to 125 genes that are exclusively linked to hyperpigmentation diseases shows striking differences (Figure 2a). For example, in the class of “Immune system and Inflammation” we find only genes involved in hypopigmentation diseases. Further, functions related to “Organelles”, are dominated by hypopigmentation genes. Indeed, these genes are linked to structural functions in lysosomes and melanosome, which are key organelles for pigment production and transport out of melanocytes. On the contrary, the class of “Chromatin organization and DNA repair” is dominated by genes in hyperpigmentation diseases (Figure 2a). This class also is the top functional association of all 49 genes linked to either mixed hypo- and hyperpigmentation disorders, pigmentation phenotype genes, or to all genes linked to multiple hyper-/hypopigmentation disorders (Figure 2b).

While the class of “Signaling” looks evenly linked to hypo- and hyperpigmentation disorder (Figure 2a), a more fine-grained analysis of subclasses shows that hypopigmentation genes are again linked to signaling functions in the immune system and programmed cell death, while genes linked to hyperpigmentation disorders in signaling are often part of the Ras-MAP kinase cascade, a key pathway leading to the activation of melanin synthesis genes (Figure 2c,d).

### 3.3. PPI Network of Pigmentation Genes and Genes Set Enrichment

Genes and proteins related to similar diseases are often part of the same cellular networks and pathways. We first obtained PPI amongst the 243 disease–gene products (Appendix A). The final network contained 169 proteins (“nodes”) and 594 interactions (“edges”) connecting these nodes (Appendix A). Hence, 70% of the pigmentation proteins are part of a connected PPI network, suggesting that proteins participate in similar or connected pathways. Of the 30% of the nodes that are unconnected, a large fraction are transcription factors and proteins in chromatin organization, which often are not part of core PPI networks as they bind to DNA. Other proteins among the unconnected in the PPI network are proteins in the extracellular space, and proteins part of cell junctions, involved in metabolism and in immune system functions.

To further investigate the PPI network properties a clustering analysis was performed. Based on MCL clustering 39 clusters were identified with the largest cluster (cluster 1) containing 30 proteins (Figure 3). The ontology terms of cluster 1 are related to DNA repair process, and interestingly DNA metabolic processes (Appendix A and Appendix A). This is expected as Xeroderma pigmentosum, Fanconi anemia, Mismatch repair cancer syndrome are the representative pigmentation disorders associated with this cluster and they are disorders know to have DNA repair mutations.

Cluster 2 (32 proteins) is linked to signaling (mainly RTK-Ras-MAP signaling) and proteins are mainly associated with hyperpigmentation disorders (Appendix A and Appendix A). Cluster 3 contains 10 proteins associated mainly to hyperpigmentation disorders and are linked to metabolisms of rRNA and telomere maintenance (Appendix A). Clusters 4 (9 proteins) and 5 (8 proteins) are mainly linked to hypopigmentation disorders and linked to organelle synthesis (cluster 4 lysosomes and cluster 5 melanosomes) (Appendix A). Indeed, cluster 4 are represented by BLOC proteins which have a role in the transport of melanin. This cluster is interestingly largely associated with Hermansky–Pudlak syndrome, with 8 out of the 9 genes associated with the pigmentation disorder. Hermansky–Pudlak syndrome is a disorder characterized by lack of colour in the hair skin and eyes as well as a platelet disorder, which also appears in Appendix A in the form of blood coagulation biological process.

### 3.4. Expansion of the Pigmentation Disease PPI Network

Next, we were interested to investigate the pigmentation disease network after extending the number of interactions by allowing disease proteins to interact not only among each other but with additional proteins from the STRING database (see Methods). Indeed, Barabasi et al. [38] showed that friends of an obese friend, even after 3 degrees of separation, has an increase of 20% in obesity compared to a randomly generated network, showing the relevance or network extension analysis. Our expanded network contains 4668 edges (Appendix A) and 452 nodes (Appendix A). Of those edges, a large fraction (2285) is between newly added proteins that are not among the pigmentation disease proteins (Figure 4).

With respect to newly added binding proteins, the majority of new edges are between hyperpigmentation proteins (752), followed by all proteins linked to multiple hyper-/hypo- and mixed pigmentation disorders (686), and edges connecting to proteins of hypopigmentation disorders (354). Likewise, more interactions are identified between proteins that are both associated with hyperpigmentation disorders compared to proteins that are both associated with hypopigmentation disorders (Figure 4). Considering that there are more almost twice as many proteins linked to hypo- compared to hyperpigmentation disorder, this confirms our findings of the first pigmentation disease network, suggesting a better PPI network coverage for hyperpigmentation proteins.

### 3.5. Protein Expression Analysis in Pigment and Non-Pigment Producing Melanocytes and Quantitative Pigmentation Network Map

Part of a function of a PPI network is determined by the expression levels of the proteins (e.g., in a cell type-specific manner). To add this type of quantitative information to the expanded network, we determined protein abundances using mass spectrometry of cell lysates of A375 (unpigmented) and FM55 (pigmented) melanocyte cells. In total, 4202 proteins were identified in A375 and 4199 proteins in FM55 cells (Appendix A). The coverage of proteins detected in each of the two cell lines is identical with 27% of the 243 pigmentation disease proteins and 40% of all 452 proteins of the expanded network detected.

We next overlaid the expanded network with the protein abundances (based on LFQ intensities) to generate a quantitative pigmentation network map for each cell line (Figure 5). While there are some quite drastic differences in expression levels comparing the two cell lines (with most remarkably, an isoform switching of different enolases; Appendix A), the expression levels of the proteins in the pigmentation network map are surprisingly similar expressed between the two cell lines. This probably suggests that pigmentation networks consists of mostly cell general expressed proteins, such as the Ras-MAPK pathway. Those networks become activated (e.g., in response to growth factors) and are activated by changes in their phosphorylation status; hence differences in pigmentation status are not necessarily a result of changes in protein abundances in the participating proteins. Indeed, only a small number of proteins of the pigmentation network are differentially expressed of >2fold. Examples are GPNMB (a transmembrane glycoprotein suggested to play a role in melanogenesis; https://www.uniprot.org/uniprotkb/Q14956/entry; accessed on 10 November 2022), NFIX (a nuclear factor 1 X-type protein), and NRP2 (Neuropilin-2, a receptor for semaphorins), which are higher expressed in FM55 cells. The proteins KRT14 (keratin filament), WRAP53 (a telomerase Cajal body protein), IRS1 (an adaptor of the insulin receptor), FANCI (Fanconi anemia group I protein; linked to both hyper- and hypopigmentation disorders), LSM14A (regulates translation and stability of mRNAs), TERF2 (telomeric repeat-binding factor), CDC45 (required for initiation of chromosomal DNA replication), and PHB (functions in mitochondrial integrity) are lower expressed in FM55 cells.

Altogether, we present a quantitative pigmentation network map based on PPI information in two melanoma cell lines. The network is visualized using Cytoscape software and is available as Appendix A.

## 4. Discussion

We have reported here that disease–gene networks show generally a clear separation between genes involved in hyperpigmentation, hypopigmentation, and mixed hyper-/hypopigmentation disorder classes (Appendix A). Within one disease class, however, there is a very fine-grained specification of disease phenotypes, where almost all genes are linked to a unique type of disease (Figure 1a). Indeed, it is not surprising that skin, a tissue composed of a large variety of cell types that contribute to maintaining homeostasis and structural integrity, and which is also exposed to the environment (e.g., UV light) has a complex genetic richness underlying the different types of pigmentation disease [39]. Furthermore, pigmentation disorder can be based on both monogenic (single gene and mainly rare genetic variants) and non-Mendelian genetics (often common gene variants).

Disease–gene products (proteins) of different disease classes show partially class-specific functionalities. Immune system and inflammation proteins are almost exclusively linked to hypopigmentation disorders. Indeed, vitiligo is as an example for a hypopigmentation disorder that has its likely cause in immune functions, where (under stress conditions) melanocytes activate the innate immune system, which activates the adaptive immune system, and ultimately causing autoimmune destruction of melanocytes [40]. Our analysis also shows that lysosomal functions are clearly linked to hypopigmentation disorders. Indeed, hypopigmentation disorders belong to a larger group of >50 lysosomal storage diseases [41]. With respect to hyperpigmentation disorders, we find Ras and MAPK pathway proteins, which belong to one of the three core pathways in regulation of melanin synthesis that lead to the activation of MITF [2].

Genes involved in metabolism are found in all classes of pigmentation disorders. Genes involved in amino acid metabolism are dominating this functional class. Defects in enzymes that catalyze the synthesis or transport of specific amino acids (e.g., tyrosin) are well-known factors in pigmentation disorders [42]. Genes involved in lipid and fatty acid metabolism are also frequently found in our list of disease genes. Indeed, recently it was shown that fatty acid metabolism is a key regulator of melanogenesis [43].

Chromatin organization, DNA repair and proteins linked to reactive oxygen species (ROS), hypoxia, and heat stress are also frequently linked to pigmentation disorders. These processes are highly connected and often become more affected with increasing age. It therefore comes as no surprise that both hypo- and hyperpigmentation disorders are suggested to be the result of age-related alterations in oxidative stress levels, DNA damage, and telomer changes (among other factors), that go in line with skin aging [44].

Following a classical network-centric approach, a main aim of this study was to understand how pathways and PPI networks are altered in pigmentation diseases. Despite a clear separation of gens into different disease phenotypes, on the networks level the disease proteins are well connected (70% of the disease proteins are part of connected PPI network). This value would probably be higher, but many genes are related to functions that are typically less covered in PPI databases, such as transcription factors, chromatin proteins, extracellular matrix proteins. Additionally, metabolic proteins are often disconnected from PPI networks as their links involve the actual metabolites.

The bioinformatics approaches used here focus on PPI networks and pathways, which present potential biases. For example, changes in posttranslational modifications (e.g., phosphorylation, methylation) and their impact on PPI network rewiring are not explicitly taken into consideration when obtaining data from general PPI databases. Furthermore, general PPI databases contain data obtained from experiments in multiple cell systems, but not necessarily in melanocytes or other skin cell types. Hence, some additional melanocyte-specific PPI and pathways might be lacking, although in general our reconstructed networks are well connected.

As the disease proteins in the skin pigmentation PPI network were remarkably interconnected, we went further and extended the PPI network into a network of 452 nodes and 4668 edges. Additional proteins that are not present in the pigmentation disorder network may still have a role in the disorder in a network neighborhood manner [45]. This network contains information from protein expression in pigmented and unpigmented melanoma cells and as such represents a quantitative pigmentation network map. However, the two melanocyte cell types considered here might not be sufficient to have a good representation of genes in melanogenesis and factors such as additional mutations might contribute to changes in protein expression levels. However, as gene expression data, also for single cells, are becoming increasingly available, the networks available from this work can be further pruned and analyzed with future data sets. The interactions of this map are either accessible through excel tables or through the Cytoscape network visualization tool. As such, we believe that this is a valuable resource that will enable the skin pigmentation research community to test experimentally new hypotheses arising from connectivities in the map.

## Figures and Tables

**Figure 1 bioengineering-10-00013-f001:**
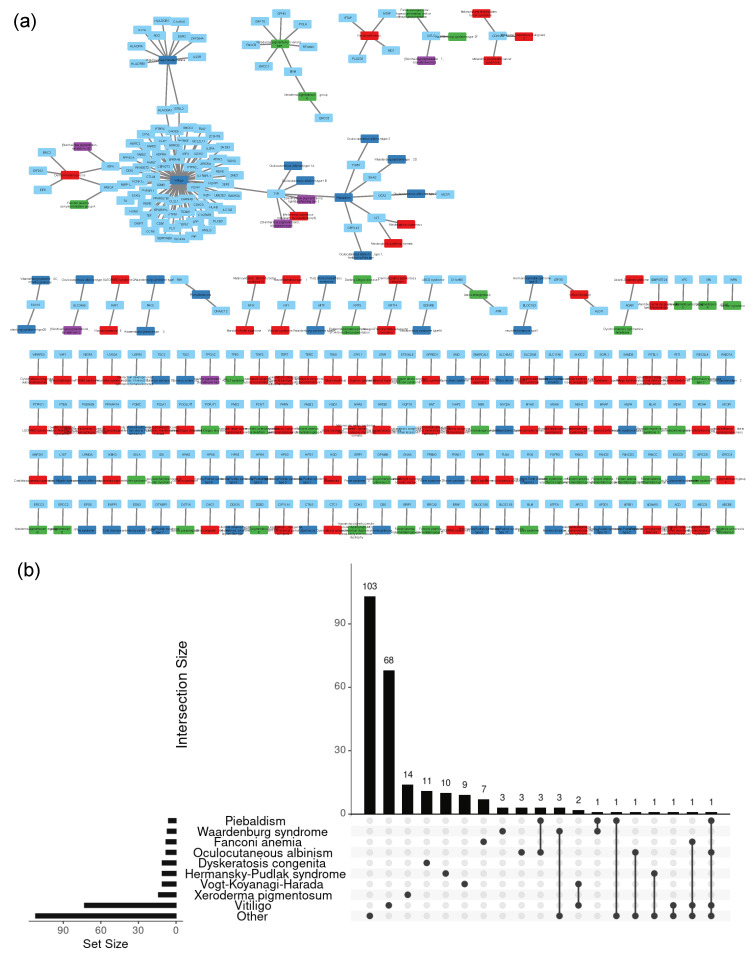
Disease–gene network of skin pigmentation disorders. (**a**) Disease–gene network of skin pigmentation disorders and pigmentation phenotypes. Light blue nodes correspond to genes that are linked to diseases colored according to sub-group: hyper-pigmentation (red), hypopigmentation (blue), mixed pigmentation (green), and pigmentation phenotype (purple). The network was represented using Cytoscape. (**b**) Upset plot, visualizing pigment disorder gene overlap. The number of gene intersections are shown in bars in the main plot. The single beads below each bar report the number of genes unique to that pigment disorder. The beads on a line show the number of genes that are common to each pigment disorders with a bead on the line.

**Figure 2 bioengineering-10-00013-f002:**
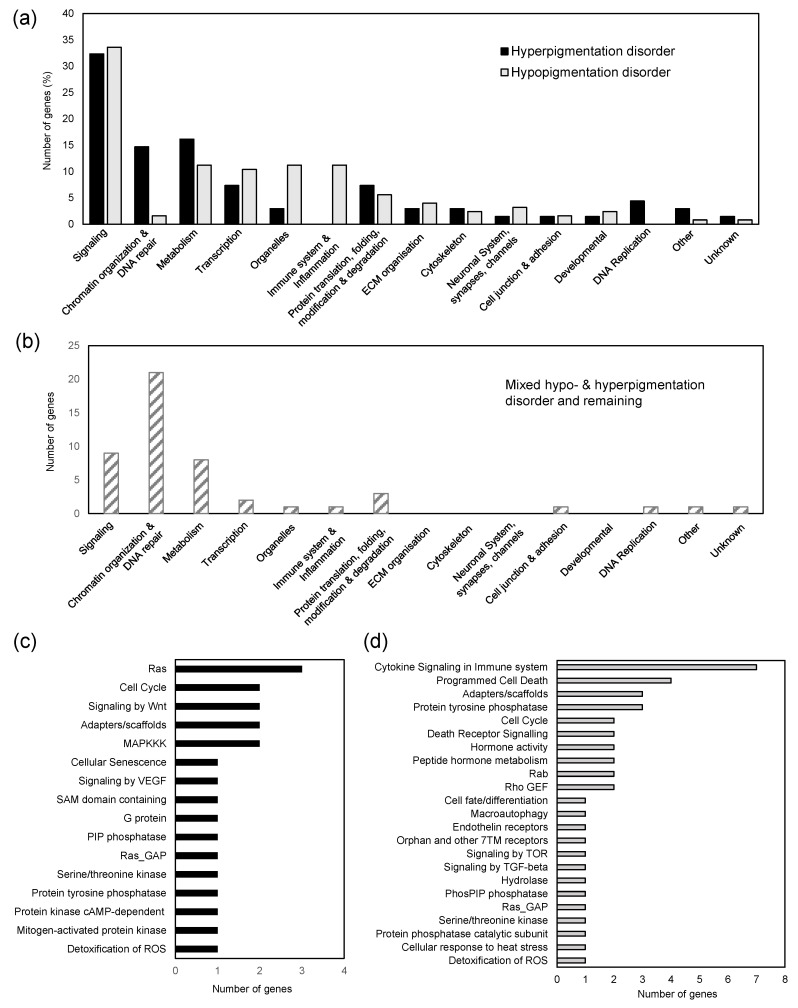
Functional analysis of pigmentation using the SysGO database (level 3). (**a**) Functional analysis comparing genes that are exclusively linked to hyperpigmentation diseases (total of 68 genes) to those that are exclusively linked to hypopigmentation disease (total of 125 genes). The number of genes is shown as percentage of total genes in each class (hyper- or hypopigmentation). (**b**) Functional analysis of genes linked to mixed hypo- and hyperpigmentation disorders and remaining genes (pigmentation phenotype genes and all genes that link to both, hyper- and hypopigmentation disorders) (total of 49 genes). (**c**) Further subclassification of hyperpigmentation genes in “signaling” into level 1 classes (total of 22 genes). Abbreviations: MAPKKK, Mitogen-activated protein kinase kinase kinase; SAM, Sterile alpha motif; PIP, Phosphatidylinositol-phosphate (**d**) Further subclassification of hypopigmentation genes in “signaling” into level 1 classes (total of 42 genes).

**Figure 3 bioengineering-10-00013-f003:**
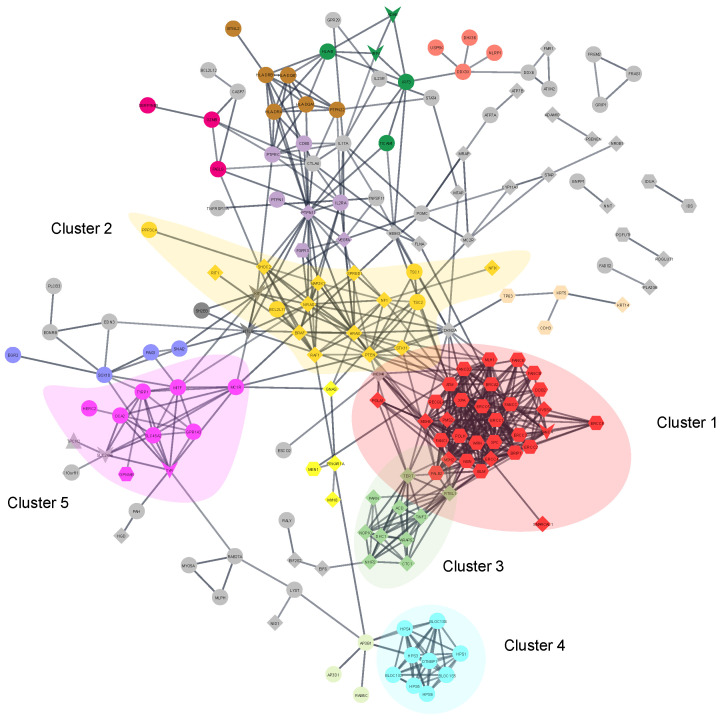
Pigment disorder PPI network and results of clustering analysis. Network among 169 pigmentation proteins based on the STRING database. Similar colors of proteins correspond to the participation in the same network cluster. The top 5 largest clusters are shaded in circles. The shapes of the proteins report the pigmentation status of the disorder associated with the protein: diamonds are hyperpigmentation, circles are hypopigmentation, hexagons are mixed pigmentation, triangles are pigmentation phenotype, and V-shape are proteins that link to multiple diseases with combined hyper-/hypo or mixed associations.

**Figure 4 bioengineering-10-00013-f004:**
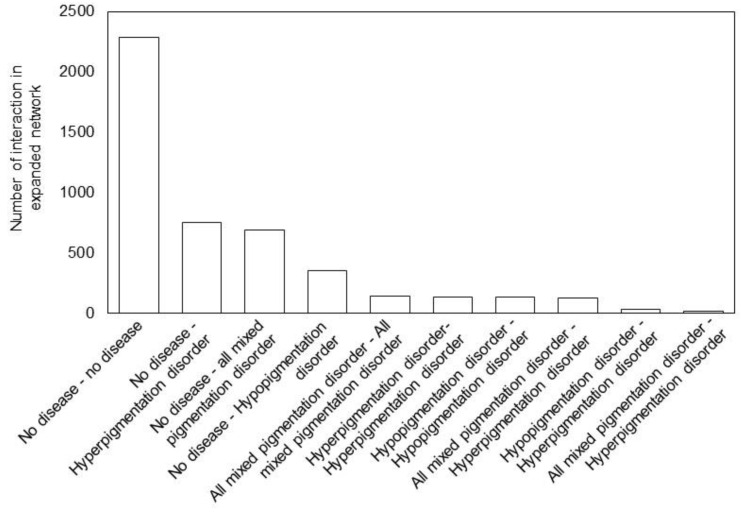
Number of interactions in the expanded network between different types of proteins. Proteins were classified into those not linked to pigmentation disorders and those linked to pigmentation disorders (Appendix A). The full list of interactions is available in Appendix A.

**Figure 5 bioengineering-10-00013-f005:**
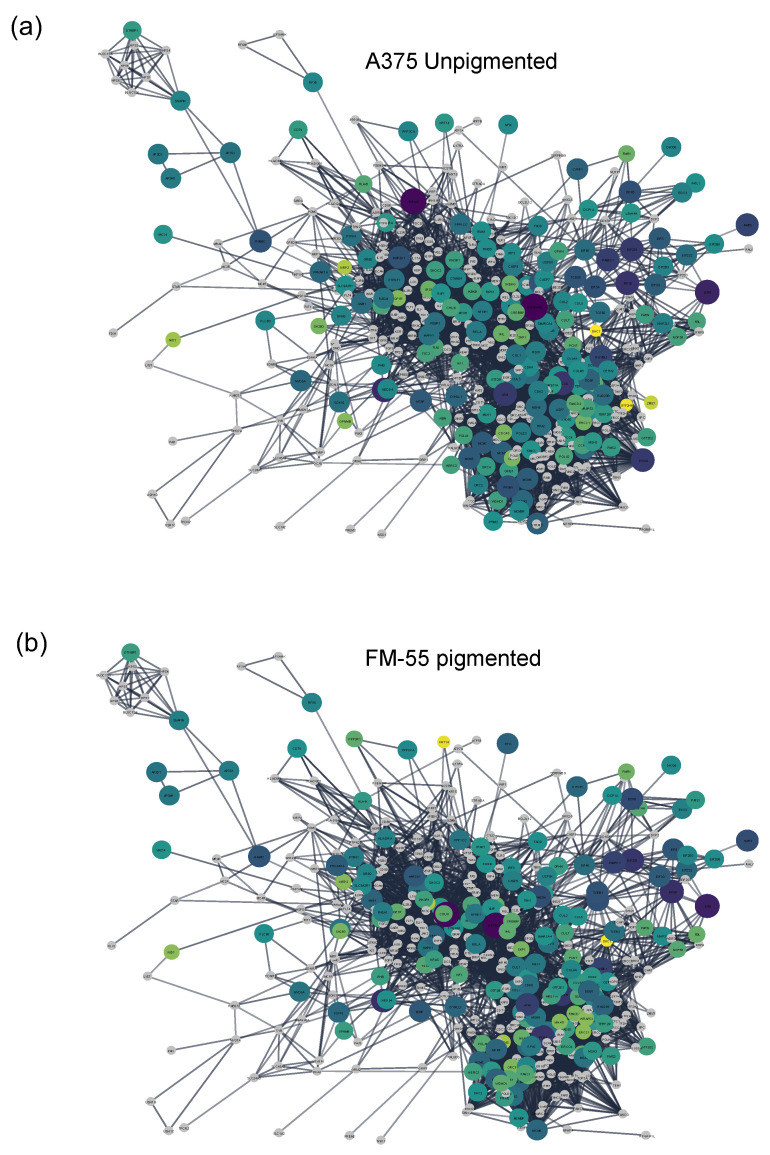
Quantitative pigmentation network map in two melanoma cell lines. (**a**) Expanded pigmentation network with protein shape and color following their expression levels in the unpigmented A375 cell line. (**b**) Expanded pigmentation network with protein shape and color following their expression levels in the FM55 pigmented cell line. The nodes are colored according to the log transformed LFQ intensity yellow = low, dark blue = high. The size of the node is also set according to the log transformed LFQ.

## Data Availability

All data generated in this work are provided in the Appendix A.

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
