# Peer review of "Disease–Gene Networks of Skin Pigmentation Disorders and Reconstruction of Protein–Protein Interaction Networks"

_bioengineering, 2022, doi:10.3390/bioengineering10010013_

Round 1

Reviewer 1 Report

1. Line 292-294 and figure 2 legends, correct the description of genes number linked with hyperpigmentation or hypopigmentation diseases.

2. Problems happen in Fig. 4

3. Please explain the method for expansion of the PPI network.

Author Response

Reviewer 1

Comment 1: “1. Line 292-294 and figure 2 legends, correct the description of genes number linked with hyperpigmentation or hypopigmentation diseases.”

Reply 1: We originally displayed the number of genes as a percentage (with respect to either all hyper or all hypopigmentation disorder genes) to be able to compare in panel (a) the functional classes for genes in hyper and hypopigmentation. We now show the real numbers (not percentages) for the remaining panels, and explain in the figure legend of panel (a) that the number of genes is in %. See also comment 25 (reviewer 2).

Comment 2: “2. Problems happen in Fig. 4”

Reply 2: We have changed figure 4 to make it better readable.

Comment  3: “3. Please explain the method for expansion of the PPI network.”

Reply 3: The methods was already included in the section 2.9. We added a bit more information. We also clarified the expanded network in section 3.4.

Reviewer 2 Report

The authors presented a very interesting paper with a new bioinformatic approach to identify proteins involved in disease-related to pigmentation disorders and link their phenotypes to protein-protein interaction networks. However, the discussion can be further enlarged to include potential bias of the methods and to discuss results for the bioinformatic and expression part, maybe taking only 2 cell lines is not sufficient to have a good representation of the genes involved in melanism, and few genes are involved in melanogenesis compared to all 20’000 human genes. Furthermore, the article seems to have been written without being reviewed by the authors, a lot of small mistakes (typing, mistake in melanogenic protein name, mixing of hypo- and hypermelanic…) that renders the reading difficult and reduces the trust in the article. For citation, normally the first article that appears on the topic should be cited.

Minor comments

L 44: Citation: Dubey, S., and Roulin, A. (2014) Evolutionary and biomedical consequences of internal melanins. Pigment Cell Melanoma Res 27, 327-338

L50: citation is missing: Slominski, A. T., Zmijewski, M. A., Skobowiat, C., Zbytek, B., Slominski, R. M., and Steketee, J. D. (2012) Sensing the environment: regulation of local and global homeostasis by the skin's neuroendocrine system. Adv Anat Embryol Cell Biol 212, v, vii, 1-115

L64:  Cite Mackintosh, J. A. (2001) The antimicrobial properties of melanocytes, melanosomes and melanin and the evolution of black skin. J. Theor. Biol. 211, 101-113 instead of 6.

L80: “TRYP2”: stands for trypsin in UniProt, do the authors mean TYRP2 / TYRP1.

L81: “TRPS* what does it stand for?

L82-101: this paragraph can be shorten, there are some repetitions (L89-92), see reference Ito, S., and Wakamatsu, K. (2008) Chemistry of mixed melanogenesis--pivotal roles of dopaquinone. Photochem Photobiol 84, 582-592. It should be mentioned that the described mechanism is for human.

L93: “TRP2” is a telomere-binding protein, do the authors mean TYRP2 or TYRP1?

L99:  SLC7A11 reference: Chintala, S., Li, W., Lamoreux, M. L., Ito, S., Wakamatsu, K., Sviderskaya, E. V., Bennett, D. C., Park, Y. M., Gahl, W. A., Huizing, M., Spritz, R. A., Ben, S., Novak, E. K., Tan, J., and Swank, R. T. (2005) Slc7a11 gene controls production of pheomelanin pigment and proliferation of cultured cells. Proc Natl Acad Sci U S A 102, 10964-10969

L113: PRKAR1A the full name can be given or the gene function.

L114-116: Oculocutaneous albinisms are multiple see the reference.

L119-120: the sentence is not clear: “…the transport of melanosomes to the skin”

L127: “it is however quite evident in monogenetic diseases that have a polygenetic phenotype” the sentence is not clear.

L132: the results of reference “25 “ can be explained to illustrate the idea.

L151: the authors could be cited in the text.

L194: What is the speed of “high speed”?

L199-208: the sentences are not clear.

L244-246: sentence not clear.

L248-252: repeat of the start of the results.

L288-292: already in materials and methods

L294: Do the authors mean “comparison of hyper- and hypopigmentation”. And why are the number of genes considered different from before.

Figure 2c and d: the number of gene can be written directly and not the % not meaningful.

            The writing of the legend is not constant.

L309-311: There are further confusion, what is what?

L359-362: the sentence is not clear.

Figure 4: not readable.

L407: LFQ what does it mean?

L411: how do the authors control for expression level to compare both cell lines?

L446: “comment” what does it mean?

L450: Do the author mean “linked”

Supplementary figure: the quality does not allow to read the name of the proteins.

Author Response

Reviewer 2

Comment 4: “The authors presented a very interesting paper with a new bioinformatic approach to identify proteins involved in disease-related to pigmentation disorders and link their phenotypes to protein-protein interaction networks. However, the discussion can be further enlarged to include potential bias of the methods and to discuss results for the bioinformatic and expression part, maybe taking only 2 cell lines is not sufficient to have a good representation of the genes involved in melanism, and few genes are involved in melanogenesis compared to all 20’000 human genes. Furthermore, the article seems to have been written without being reviewed by the authors, a lot of small mistakes (typing, mistake in melanogenic protein name, mixing of hypo- and hypermelanic…) that renders the reading difficult and reduces the trust in the article. For citation, normally the first article that appears on the topic should be cited.”

Reply 4: Thank you for the comments. We have expanded the discussion and carefully checked for any remaining typos, and improved citations.

Minor comments

Comment 5: “L 44: Citation: Dubey, S., and Roulin, A. (2014) Evolutionary and biomedical consequences of internal melanins. Pigment Cell Melanoma Res 27, 327-338”

Reply 5: We have added this reference.

Comment 6: “L50: citation is missing: Slominski, A. T., Zmijewski, M. A., Skobowiat, C., Zbytek, B., Slominski, R. M., and Steketee, J. D. (2012) Sensing the environment: regulation of local and global homeostasis by the skin's neuroendocrine system. Adv Anat Embryol Cell Biol 212, v, vii, 1-115”

Reply 6: We have added this reference.

Comment 7: “L64:  Cite Mackintosh, J. A. (2001) The antimicrobial properties of melanocytes, melanosomes and melanin and the evolution of black skin. J. Theor. Biol. 211, 101-113 instead of 6.”

Reply 7: We have replaced the reference.

Comment 8: “L80: “TRYP2”: stands for trypsin in UniProt, do the authors mean TYRP2 / TYRP1.”

Reply 8: We apologize for the confusion in this sentence. We now correctly define the three key enzymes: tyrosinase (TYR), dopachrome tautomerase (TYRP2), and tyrosine-related protein (TYRP1).

Comment 9: “L81: “TRPS* what does it stand for?”

Reply 9: We apologize for the confusion in this sentence. We now correctly define the three key enzymes: tyrosinase (TYR), dopachrome tautomerase (TYRP2), and tyrosine-related protein (TYRP1).

Comment 10: “L82-101: this paragraph can be shorten, there are some repetitions (L89-92), see reference Ito, S., and Wakamatsu, K. (2008) Chemistry of mixed melanogenesis--pivotal roles of dopaquinone. Photochem Photobiol 84, 582-592. It should be mentioned that the described mechanism is for human.”

Reply 10: We have shortened the paragraph and added the reference.

Comment 11: “L93: “TRP2” is a telomere-binding protein, do the authors mean TYRP2 or TYRP1?”

Reply 11: We apologize for the confusion in this sentence. We now correctly define the three key enzymes: tyrosinase (TYR), dopachrome tautomerase (TYRP2), and tyrosine-related protein (TYRP1).

Comment 12: “L99:  SLC7A11 reference: Chintala, S., Li, W., Lamoreux, M. L., Ito, S., Wakamatsu, K., Sviderskaya, E. V., Bennett, D. C., Park, Y. M., Gahl, W. A., Huizing, M., Spritz, R. A., Ben, S., Novak, E. K., Tan, J., and Swank, R. T. (2005) Slc7a11 gene controls production of pheomelanin pigment and proliferation of cultured cells. Proc Natl Acad Sci U S A 102, 10964-10969”

Reply 12: We have added this reference.

Comment 13: “L113: PRKAR1A the full name can be given or the gene function.”

Reply 13: We added the full name.

Comment 14: “L114-116: Oculocutaneous albinisms are multiple see the reference.”

Reply 14: We mention now that albinism can manifest as either syndromic or non-syndromic phenotypes.

Comment 15: “L119-120: the sentence is not clear: “…the transport of melanosomes to the skin””

Reply 15: We have corrected it.

Comment 16: “L127: “it is however quite evident in monogenetic diseases that have a polygenetic phenotype” the sentence is not clear.”

Reply 16: This was indeed written in a misleading way. We have corrected the sentence.

Comment 17: “L132: the results of reference “25 “ can be explained to illustrate the idea.”

Reply 17: We have done this now. “For example, genes involved in monogenic RD affect core vision related functions (e.g. phototransduction, retinal recycling), while genes involved in complex and age-related RD are linked to genral tissue homeostasis processes (e.g. extracellular matrix remodel-ling, inflammation) (26).”

Comment 18: “L151: the authors could be cited in the text.”

Reply 18: We have added “Baxter et al” to the text.

Comment 19: “L194: What is the speed of “high speed”?”

Reply 19: We have added this information now.

Comment 20: “L199-208: the sentences are not clear.”

Reply 20: We have corrected those sentences.

Comment 21: “L244-246: sentence not clear.”

Reply 21: We have corrected the sentence.

Comment 22: “L248-252: repeat of the start of the results.”

Reply 22: We have amended this part.

Comment 23: “L288-292: already in materials and methods”

Reply 23: We have deleted this part and refer to the Methos section.

Comment 24: “L294: Do the authors mean “comparison of hyper- and hypopigmentation”. And why are the number of genes considered different from before.”

Reply 24: ‘We have corrected hyper’ to ‘hypo’ in two instances. The numbers are different as we are looking at genes that exclusively either link to hypo or hyper pigmentation.

Comment 25: “Figure 2c and d: the number of gene can be written directly and not the % not meaningful.”

Reply 25: We have done this.

Comment 26: “The writing of the legend is not constant.”

Reply 26: We have rewritten the legend.

Comment 27: “L309-311: There are further confusion, what is what?”

Reply 27: We have corrected this (hypo instead of hyper in two instances (legend fig 2.)

Comment 28: “L359-362: the sentence is not clear.”

Reply 28: We have made this sentence clearer.

Comment 29: “Figure 4: not readable.”

Reply 29: We have increased the font size of Figure 4.

Comment 30: “L407: LFQ what does it mean?”

Reply 30: LFQ is “label free quantitation”. It was already defined in section 2.8.

Comment 31: “L411: how do the authors control for expression level to compare both cell lines?”

Reply 31: Protein expression levels are compared at the level of LFQ (label free quantification) intensities. LFQ intensities are calculated using MaxQuant (see Methods).

Comment 32: “L446: “comment” what does it mean?”

Reply 32: Sorry, it should have been “common”. We have corrected it.

Comment 33: “L450: Do the author mean “linked””

Reply 33: Yes, sorry, “linked”. We have corrected it.

Comment 34: “Supplementary figure: the quality does not allow to read the name of the proteins.”

Reply 34: We have increased the resolution.

Reviewer 3 Report

This paper represents an important step in our understanding of the interplay between pigmentation and cellular biology beyond simple UV protection. The paper could be improved in the following manner.

1)On line 66 you mention the role of melanin as an antibacterial. This needs to be cited as this is not a common role.

2) Line 116, albinism is not the result of only TYR loss, there are other forms of albinism. statement should be amended. Loss of tyrosine occurs over life? or do you mean loss of tyrosinase?

3) Figure 2 legend has errors. hyperpigmentation is listed instead of hypopigmentation.

4) Since only one pigmented and one non-pigmented line was used it is hard to draw any firm conclusions. I think it is best to minimize the emphasis on these cells as other explanations for difference might relate to donor, other mutations, etc.

Author Response

Reviewer 3

This paper represents an important step in our understanding of the interplay between pigmentation and cellular biology beyond simple UV protection. The paper could be improved in the following manner.

Comment 35: “1)On line 66 you mention the role of melanin as an antibacterial. This needs to be cited as this is not a common role.”

Reply 35: We put this citation now (Ref 8).

Comment 36: “2) Line 116, albinism is not the result of only TYR loss, there are other forms of albinism. statement should be amended. Loss of tyrosine occurs over life? or do you mean loss of tyrosinase?”

Reply 36: We added now that tyrosinase loss is an example for the development of albinism. We also clarify now that it is a loss of tyrosine because of oxidative posttranslational modifications.

Comment 37: “3) Figure 2 legend has errors. hyperpigmentation is listed instead of hypopigmentation.”

Reply 37: We have corrected this.

Comment 38: “4) Since only one pigmented and one non-pigmented line was used it is hard to draw any firm conclusions. I think it is best to minimize the emphasis on these cells as other explanations for difference might relate to donor, other mutations, etc.”

Reply 38: We have included this now in the discussion.
